# Preventing Postpartum Depression in the Early Postpartum Period Using an App-Based Cognitive Behavioral Therapy Program: A Pilot Randomized Controlled Study

**DOI:** 10.3390/ijerph192416824

**Published:** 2022-12-15

**Authors:** Xiaoli Qin, Chunfeng Liu, Wei Zhu, Yan Chen, Yudong Wang

**Affiliations:** 1The International Peace Maternity and Child Health Hospital, School of Medicine, Shanghai Jiao Tong University, Shanghai 200030, China; 2Shanghai Key Laboratory of Embryo Original Diseases, Shanghai 200030, China; 3School of Electrical and Information Engineering, The University of Sydney, Sydney 2006, Australia

**Keywords:** digital mental health, cognitive behavioral therapy, postpartum depression, postpartum anxiety, depression prevention

## Abstract

A large proportion of women experience depression during the postpartum period. Few studies have investigated the use of mobile technology to prevent postpartum depression in women. This study investigated the preliminary effectiveness of the CareMom program, a new app-based cognitive behavioral therapy program, on reducing the depressive symptoms of mothers during the very early postpartum period via a pilot randomized controlled study. The participants were recruited during birth hospitalization (within 3 days after giving birth) and randomized to the waiting-list control and CareMom groups. Over the four-week intervention period, the CareMom group was required to complete 28 daily challenges via CareMom. The depressive (via EPDS) and anxiety (via GAD-7) levels of participants were measured at baseline and every 7 days postbaseline for 4 weeks. A total of 112 eligible participants were randomly allocated to the two groups (CareMom: n = 57; control: n = 55). At week 4, the CareMom group achieved a significantly lower EPDS score than the control group at week 4 (*p* = 0.037). In addition, the EPDS (*p* < 0.001) scores of the CareMom group were significantly lower than the baseline values. However, the control group did not show any significant reduction in this measure. No significant reduction of GAD-7 scores was observed for CareMom and control groups at week 4. This study provides preliminary evidence of the effectiveness of CareMom in reducing depressive symptoms in the general postpartum population during the very early postpartum period.

## 1. Introduction

Due to changes in physiological and psychosocial aspects, many women experience depressive symptoms during the postpartum period [1]. There is currently no consensus on the definition of the postpartum period among clinicians and researchers in the mental health domain. In the fifth edition of the Diagnostic and Statistical Manual of Mental Disorders, the postpartum period is defined as being within 4 weeks postpartum [2]. However, considering many depressive episodes occur after 4 weeks postpartum, some researchers and clinicians argue that the timeframe of the postpartum period should be increased to 6 months [3] or 1 year after delivery [4].

Within the first three days after childbirth, more than 50% of mothers experience postpartum blues, which are the mildest psychological discomfort with symptoms of crying, sadness, and unstable emotions [5]. Generally, postpartum blues do not require treatment, and many mothers with postpartum blues will recover within weeks [6]. For some mothers, this psychological discomfort then develops into postpartum depression [7]. According to previous research, up to 15% of mothers experience postpartum depression [8]. Postpartum depression not only influences the mother but also the family and the child’s development [9,10]. Given the negative impacts of postpartum depression, we assume that providing psychological support and education during the very early postpartum period (e.g., within the first month postpartum) might be helpful in preventing the development of postpartum blues into postpartum depression. The recommendation statement from the US Preventive Services Task Force (USPSTF) also confirmed that the immediate postpartum period would be a reasonable time to offer preventive interventions for postpartum depression [11].

According to a USPSTF statement, psychological interventions such as interpersonal therapy [12] and cognitive behavioral therapy (CBT) [13] have been found to be effective in preventing postpartum depression [11]. CBT is one of the most recommended evidence-based methods for treating and preventing postpartum depression [14]. The core theoretical framework of CBT states that people’s dysfunctional patterns of cognition are the causes of their emotional distress and maladaptive behaviors [15]. To improve emotional and behavioral states, people must change their dysfunctional cognitive patterns. CBT provides people with methods and tools to evaluate, challenge, and modify dysfunctional cognitive patterns [16]. 

Traditionally, CBT is delivered by health professionals in an individual or group face-to-face format [13]. Over the past decade, with the development of the Internet and mobile technologies, multiple CBT-based online platforms have been proposed for postpartum mothers. Most platforms focus on the treatment of postpartum depression [17,18,19]. Very few online platforms have been developed to prevent postpartum depression among mothers. Fonseca et al. have proposed a web-based platform to prevent postpartum depression [20,21]. Their preliminary study results showed that women in the intervention group presented with a larger decrease in depressive and anxiety symptoms [21]. The study participants were women who presented with risk factors or early onset postpartum depression, and the study was conducted from the second month postpartum. 

Different types of prevention have been applied in the mental health domain [22]. Universal prevention is a type of prevention aimed at the entire population. Selective prevention is aimed at the high-risk population, and indicated prevention is aimed at the population with emerging symptoms. The study conducted by Fonseca et al. applied selective and indicated prevention strategies [21]. However, few studies have investigated the effectiveness of online platforms in the universal prevention of postpartum depression. 

One large universal prevention study showed that an app-based CBT program was not effective in preventing postpartum depression in the general postpartum population [23]. However, the intervention of the study was conducted in pregnant women in the second trimester rather than the early postpartum period. Another study investigated the effect of a CBT-based chatbot program on depression symptoms in the general postpartum population during the early postpartum period [24]. Mothers with and without clinical depressive symptoms were included in the study and provided a chatbot program since their birth hospitalization. However, compared to the control group, depression scores were not significantly reduced in the six-week postpartum period [24]. 

In this study, we proposed a new app-based program called CareMom that delivers CBT contents to prevent postpartum depression in the general postpartum population in the very early postpartum period. As CareMom has not been tested via a controlled study previously, we defined this research as a pilot study to conduct preliminary evaluations of the program [25]. The aims of this pilot study were (1) to explore the preliminary effectiveness of the CareMom program on the universal prevention of postpartum depression via a randomized control study, and (2) to evaluate the acceptability of the CareMom program. In this study, postpartum women were recruited during their birth hospitalization and randomized to use the CareMom program with the usual postpartum care or to receive the usual postpartum care only. Participants’ depressive and anxiety symptoms were measured using valid self-report surveys. Depressive symptom reduction is the main outcome for evaluating the effectiveness of universal prevention [26]. We hypothesized that using the CareMom program would reduce participants’ depressive symptoms and that participants would consider CareMom an acceptable program.

## 2. Methods

### 2.1. Participants

The study was conducted in a public hospital located in Shanghai, China. The participants were Chinese-speaking women who delivered a live-born neonate in the hospital and were recruited during their birth hospitalization. The key criteria for inclusion were as follows: (1) 0 days to 3 days after delivery; (2) the mother was not diagnosed with any mental disorders at any time; (3) the mother was not experiencing severe depressive symptoms during the recruitment, which was evaluated by an EPDS score less than 16; (4) the baby was not diagnosed with any severe illnesses; (5) the mother did not experience severe physical illnesses during the recruitment; (6) the mother did not have drug or alcohol issues over the past 12 months; (7) the mother was not undergoing any kinds of psychological services or treatments; (8) the mother owned a smartphone; and (9) the mother was available to independently engage with the program for 4 weeks.

In this study, to detect a medium effect size (d = 0.65) for a two-tailed independent *t*-test at a significance level of 5% with a power of 80%, at least 39 participants were required for each group [27]. Considering the expected dropout rate in our study (30%) [28], at least 112 participants were recruited for this pilot study. G* Power software was used for the sample size calculation.

This study was approved by the ethics committee of The International Peace Maternity and Child Health Hospital (Reference Number: GKLW 2021-12). All the participants signed a consent form during recruitment. There was no modification of the study after its commencement. All the participants received 200 Chinese Yuan (approximately 27.7 US dollars) as a reward.

### 2.2. Measurements

We used three surveys to evaluate participants’ depressive and anxiety symptoms and the acceptability of the CareMom program.

EPDS: The Edinburgh Postnatal Depression Scale (EPDS) is a valid self-report survey to measure postpartum depressive symptoms over the past seven days [29]. The validity of the EPDS in measuring postpartum depression after birth has been examined in previous studies [30]. The survey contains 10 questions, each of which can be scored from 0 to 3. The highest score on the EPDS is 30, and a higher score suggests more severe depressive symptoms. In this study, we used the Chinese version of EPDS [31]. Generally, for Chinese women, EPDS scores greater than 9 indicate a high likelihood of depression [31] and scores greater than 15 indicate severe depression [32]. 

GAD-7: The 7-item Generalized Anxiety Disorder (GAD-7) is a valid self-report survey to measure anxiety symptoms within the past 14 days [33]. The GAD-7 includes seven questions, each of which can be scored from 0 to 3. The total score ranges from 0 to 21, with higher scores indicating more severe symptoms of anxiety. The score range for mild anxiety is 4–9, and a score greater than 14 indicates severe anxiety. The Chinese version of GAD-7 was used in this study [34]. 

Acceptability: We used mixed-format questions to assess the acceptability and usability of the CareMom program [35]. First, participants were asked to evaluate their overall satisfaction with the program via a 5-point Likert question, with 1 being the lowest and 5 being the highest. Participants were also required to rate the extent to which they would recommend the program to their friends if they were in the postpartum period. The next two questions asked to what extent they thought the content delivered in the program was related to their daily life (1 represented “not at all”, 5 represented “a lot”) and to what extent they would apply their learning from the program to their daily life (1 represents “not at all”, 5 represents “a lot”). Participants were also asked to select the most helpful component within the program and report any technical issues they experienced. The last question of the survey was a free-text question, and participants could leave comments about the program.

### 2.3. CareMom Program

Technically, the CareMom program is a WeChat mini program in Chinese language, and users can open the program by scanning a QR code in the WeChat application. The program was designed and developed by Shanghai Thoven Technology Co., Ltd., Shanghai, China. Psychologists and psychiatrists from Shanghai Mental Health Center guided the structural design of the program and contents of the daily challenges. 

The contents of the program mainly contain psychoeducation and cognitive restructuring. As the core strategy of CBT, the effectiveness of cognitive restructuring has been tested for depression [36]. Cognitive restructuring was often combined with behavioral strategies to provide the intervention [36]. However, considering the logistical issues of applying behavioral interventions at the early postpartum period, the CareMom program does not include the contents related to behavioral interventions. 

Third-wave CBTs, such as mindfulness-based cognitive therapy (MBCT), have also been suggested for the treatment and prevention of depression [37]. However, due to the limited period of the intervention program, we did not include other psychological interventions into the CareMom program.

CareMom comprises two main components: daily challenges and mood management. These components are briefly described in the following sections.

Daily challenge component: The daily challenge component includes 28 challenges, and users can complete one of these challenges each day (see Figure 1). In the first 14 challenges, each includes a video and a few quiz questions. The length of the videos is 2 to 4 min, and the videos cover CBT topics related to postpartum depression, such as human emotions, different kinds of cognitive distortions, and methods of challenging cognitive distortions. Quiz questions are designed to test users’ understanding of the video content. The last 14 challenges contain only the quiz questions. Table 1 describes the structure and objectives of the daily challenges. The program automatically releases one challenge each day as the user activates her account. If the user misses a daily challenge, she can complete that challenge in the later days.

Mood management component: This component is designed for users to record their daily moods and reflect on the events and thoughts related to the mood. When the user logs into the program for the first time in a day, she will be asked to rate her overall daily mood from five options: very good, good, neutral, bad, and very bad (see Figure 2).

If the user selects negative emotions (bad and very bad), the program will navigate her to reflect on the events and thoughts related to her negative emotions, and it will guide her to challenge and reconstruct her negative thoughts. This process of identifying, challenging, and reconstructing negative thoughts is one of the core concepts of the CBT framework to improve people’s emotional and behavioral states [38]. When users select positive (good and very good) or neutral emotions, the program encourages them to record their positive events on the current day. This feature is inspired by the theory of positive psychology [39].

Users can review their mood records via the mood calendar functionality (see Figure 3). By clicking on the emoji on the calendar, the user can check the reflections associated with the emotion. To motivate users to complete daily challenges and actively track their mood, gamification factors were incorporated into the design of the program. Specifically, 10 points were granted to the user as a reward if they completed a daily challenge, and 5 points were granted if a mood record was completed. Repeating a completed daily challenge will attract no points, and users can only track their mood once each day. The data records of the users’ operations were automatically collected by the program.

### 2.4. Study Design and Procedures

The participants were recruited during hospitalization after delivery. The researchers, who were doctors of the hospital, introduced the study procedures and requirements to mothers in postnatal wards. If a mother consented to participate, she completed the consent and personal information forms. The personal information form included questions related to the study’s inclusion criteria. The administrator of this study then added the WeChat contact of the potential participants to ask them to complete the baseline clinical measures, which included the EPDS and GAD-7 surveys. These measures were completed via a Chinese online survey platform called Wenjuanxing (https://www.wjx.cn/, 13 December 2022). After reviewing the personal information and baseline measures of potential participants, ineligible participants were excluded from the study. The administrator then allocated eligible participants to the waiting-list control group or CareMom group using a computer-generated random number (see Figure 4). The random numbers were between 0 and 1. If the number was greater than 0.5, the participant was allocated to the CareMom group. The participant was allocated to the control group if the random number was less than 0.5.

During the intervention period, the control group was required to complete an online measure every seven days from the day of recruitment for four weeks, and the measure included EPDS and GAD-7 surveys. No additional care was provided to the control group, apart from the usual postpartum care. Participants in the CareMom group were required to complete all daily challenges four weeks after recruitment. Similar to the control group, the CareMom group also needed to complete an EPDS and GAD-7 survey every seven days for four weeks. After completing the measure in week 4, the CareMom group was asked to complete the online accessibility survey. 

All the surveys (including the EPDS, GAD-7, and the acceptability survey) in this study were collected via the Wenjuanxing online survey platform. The study administrator sent the URL of the surveys to the participants via the WeChat mobile application. During the study, the study administrator checked the completion rate of the daily challenges of each participants every morning. If a participant did not complete any daily challenges for three consecutive days, the study administrator would manually send a WeChat message to the participant to remind her to log in to the program. Although the administrator was not blinded to the group allocation, the administrator’s messages communicated via WeChat were standardized to all participants, and the administrator did not provide participants with any additional information that was not related to the study.

## 3. Results

Figure 4 shows the participants’ flow through the study. A total of 116 mothers provided voluntary consent to participate in the study between November and December 2021. Among these potential participants, 112 met the inclusion criteria. As a result, 57 and 55 participants were randomly allocated to the CareMom and control groups, respectively.

### 3.1. Demographic Information

Table 2 shows the basic demographic information and baseline scores of the clinical measures for the entire study sample (N = 112). On average, the participants were 31.9 years old (SD = 3.62). Approximately three-quarters of the participants were first-time mothers, and 46.4% experienced caesarean delivery. 

At baseline, the mean EPDS scores of the CareMom and control group were 4.54 (SD = 3.24) and 5.42 (SD = 3.59), respectively. The mean GAD-7 score of the CareMom group was 1.74 (SD = 2.11) and that of the control group was 2.26 (SD = 2.82). According to the independent-samples *t*-test, there was no significant difference between the two groups in the EPDS and GAD-7 scores (*p* > 0.05).

### 3.2. Attrition

Of the randomized participants, 93.8% (105/112) completed the EPDS and GAD-7 surveys in week 4, which means that the overall attrition rate in this study was 6.2%. Specifically, the attrition rates in the CareMom and control groups were 8.8% (5/57) and 3.6% (2/55), respectively. According to the chi-square test, there was no significant difference in the attrition rate between the two groups (*p* > 0.05).

### 3.3. Depression and Anxiety Symptoms

In this study, we assessed participants’ depression and anxiety symptoms using the EPDS and GAD-7 measures weekly for four weeks. Table 3 shows the mean EPDS and GAD-7 scores of both groups at baseline, and from week 1 to week 4. The *p* values reported in Table 3 are adjusted *p* values after applying the Bonferroni corrections for multiple comparisons. 

At week 4, we found that the CareMom group had a significantly lower EPDS score than the control group (CareMom: mean = 2.71 [SD = 2.75], control: mean = 4.55 [SD = 3.87]; independent-samples *t*-test: *p* = 0.037). In addition, after 4 weeks of intervention, the CareMom group had significantly reduced their average EPDS score from 4.58 to 2.71 (paired-samples *t*-test: *p* < 0.001), whereas the EPDS reduction of the control group was not significant (paired-samples *t*-test: *p* > 0.05). 

In terms of the GAD-7 score, at week 4, the GAD-7 score of the CareMom group was not significantly different from that of the control group (independent-samples *t*-test: *p* > 0.05). Moreover, compared to the baseline score, the CareMom and control groups both achieved a lower mean score in week 4, however the reduction was not statistically significant. 

### 3.4. Acceptability and Usability

In the CareMom group, 92.3% (48/52) of the participants who completed the study completed the acceptability survey. As shown in Table 4, the overall satisfaction level with the CareMom program was high, at 4.58 out of 5 (SD = 0.74). In addition, the participants endorsed recommending the program to their friends, with a mean score of 4.54 out of 5 (SD = 0.80). The results also showed that participants felt that the program content was related to their everyday life (mean = 4.44/5, SD = 0.62), and that they would like to apply the content they learned to their everyday life (mean = 4.44/5, SD = 0.58).

From a technical perspective, 10.4% (5/48) of the participants reported that they experienced technical issues when interacting with the program at certain points. The top-reported technical issues include app crashing and video freezing.

The engagement of the CareMom group in daily challenges is illustrated in Figure 5. More than 90% (48/52) of the participants completed all 28 daily challenges, and only two participants (3.8%) completed less than half of the challenges. On average, the participants spent 4.46 min (SD = 8.28) completing a daily challenge. Specifically, for the challenges of the first 14 days that contained video and quiz questions, participants spent 7.48 min (SD = 11.18) to complete each of them. For the last 14 daily challenges that contained only quiz questions, participants needed 1.89 min (SD = 2.53) to complete one challenge.

## 4. Discussion

### 4.1. Preliminary Effectiveness of CareMom

In this study, we introduced CareMom, a new app-based CBT program comprising 28 daily challenges and a mood management component. This pilot study aimed to investigate the preliminary effectiveness of the CareMom program on the universal prevention of postpartum depression over a four-week period. The main outcome of the intervention was the reduction of depressive and anxiety symptoms [26]. We hypothesized that using the CareMom program would reduce depressive symptoms.

The results of this study confirmed this hypothesis. The CareMom group achieved a significantly lower depression score than the control group at week 4. Furthermore, after using the CareMom program for 4 weeks, the participants in the CareMom group had significantly reduced depression scores compared with their baseline scores. In contrast, the control group did not show any significant reduction in the depression scores. These results demonstrate the preliminary effectiveness of CareMom in reducing depressive symptoms in mothers during the early postpartum period. Although the study participants did not present clinical levels of depressive symptoms, the reduction of depressive symptoms may still have a protective effect on the development of clinical postpartum depression [21].

The CareMom group did not significantly reduce their anxiety scores, and the anxiety scores between the CareMom and control groups were not significantly different at week 4. These results indicate that the effectiveness of CareMom in reducing anxiety symptoms was not evident. Given that the content and features of the CareMom program were mainly developed to prevent postpartum depression, this result is unsurprising. 

Although previous studies have investigated the effectiveness of online CBT programs in preventing postpartum depression, they have mainly focused on women who had already shown clinical symptoms or were at high risk [21,40]. However, little research has been devoted to preventing postpartum depression in the general postpartum population. Nishi et al. conducted a large universal prevention study to test the effectiveness of an app-based CBT program on preventing postpartum depression in the general postpartum population [23]. The results of the trial showed that the program did not significantly reduce the depressive symptoms of the participants in the postpartum period, compared to the control group. One of the main differences between the trial conducted by Nishi et al. and the CareMom trial was that the intervention of CareMom program was provided during the very early postpartum period. Therefore, we believe that the time of delivering the intervention might be important to the result. 

Suharwardy et al. conducted a study that aimed to improve the psychological health of new mothers during their early postpartum period via a CBT chatbot program [24]. Although their study did not show significant results, it still revealed the importance and potential benefits of an app-based CBT program during the early postpartum period. In contrast to the chatbot, the CareMom program mainly delivered the interventional contents via videos, which might have made it easier for mothers to accept the cognitive restructuring techniques.

To our knowledge, the present study is the first pilot study that has aimed to use an app-based CBT program to prevent postpartum depression in the general postpartum population during the very early postpartum period. However, due to the pilot nature and small sample size of the present study, the effectiveness of the CareMom program on the universal prevention of postpartum depression should be further investigated with a larger sample size.

### 4.2. Acceptability of CareMom

CBT, as an evidence-based method for improving people’s psychological health, is generally delivered by health professionals in a face-to-face format. However, in the early postpartum period, it is not convenient for mothers to honor face-to-face appointments for various reasons, such as personal health and commuting issues. Therefore, app-based programs have great potential to help new mothers improve their psychological health more conveniently. 

Although app-based programs provide accessible and scalable mental health interventions, the low user engagement of the programs is still an issue [41]. According to a review of 93 real-world mental health applications, the 30 day retention rate of the reviewed applications was only 3.3% [42]. Therefore, improving user engagement is significant to app-based mental health programs. In the present study, more than 90% of the participants completed the 28 day study. This result indicated the user engagement of the CareMom program was good. We believe that improving acceptability was a key aspect to enhance the user engagement. 

Compared to other studies [23], several other factors might also contribute to the high completion rate of this study. First, the research study was initially introduced by doctors to the participants during the recruitment, and the trust between the doctors and the participant might improve the motivation of the participants. Second, the study administrator manually sent WeChat reminders to the participants when participants did not complete the required tasks, and the reminders might also enhance the engagement of the participants. 

According to the acceptability survey results of the study, participants had high satisfaction scores for the CareMom program. This confirmed our hypothesis that CareMom was acceptable to the participants. We assumed that two main factors contributed to the high satisfaction scores. First, as shown through the two questions of the acceptability survey, participants felt that the content delivered by CareMom was very much related to their daily lives, and they were willing to apply the content to their daily lives. Second, the participants did not need to spend a long time on the program. On average, they spent around four and half minutes per day, or about half an hour per week, to finish the required challenges. This was shorter than a typical face-to-face CBT consultation or session in other app-based CBT programs, which were generally 45–60 min [13,19]. Thus, we believe that designing CBT content that is close to users’ personal experiences and delivering it in a simple way might be important factors to achieve better acceptability of an app-based CBT program. However, caution is needed to interpret the results of the program acceptability and the completion rate, as participants joined the study voluntarily and self-selection bias might affect the results [43].

Although in this study a four-week intervention led to a measurable reduction in participants’ depression scores, future studies would be beneficial for investigating the proper length of CareMom intervention that could provide sustained improvement in the psychological health of mothers. Furthermore, more research is needed to explore how each component influences participants’ depression scores, and which component is the most useful. In addition, due to the study design, noncompleters were not followed up in this pilot study. However, we believe that future studies should explore the reasons for dropping out of the program. Lastly, although the user engagement of the program was good, we still need to investigate the user engagement of the program in the real world with more users.

### 4.3. Study Limitations

Our study has several limitations. First, this study was conducted in a hospital in a large city, and the sociodemographic characteristics of the participants might be different from those of mothers in smaller cities or rural areas. In addition, in the present study, the sociodemographic characteristics of the participants were not fully collected or analyzed. Further studies should be conducted among postpartum populations with different sociodemographic characteristics to investigate the effectiveness of CareMom. Second, participants’ depressive symptoms were only assessed in the first month postpartum. However, postpartum depression can occur within 6 months after delivery [3]. Further investigations covering a wider range of postpartum periods would be helpful. The third limitation is the design of the control group. In this study, we did not include a control group with placebo application. The placebo effect should be considered in psychological interventions [44]. Therefore, further studies with placebo controls will provide more evidence regarding the effectiveness of the CareMom program. Fourth, the lack of information about participants’ previous clinical mental health history and the history of previously received therapy were another limitation of this study. These limitations should be considered when interpreting the results of this study. Lastly, as we defined the present study as a pilot study, the trial was not registered, and the protocol of the trial has not been published previously. We advise readers to carefully assess the validity of any potential explicit or implicit claims related to the outcomes of the study.

## 5. Conclusions

This study demonstrates that an app-based CBT program has the potential to reduce depressive symptoms in the general postpartum population during the very early postpartum period. Importantly, this study also showed that the CareMom program is a practical and helpful tool for delivering CBT to postpartum mothers.

## Figures and Tables

**Figure 1 ijerph-19-16824-f001:**
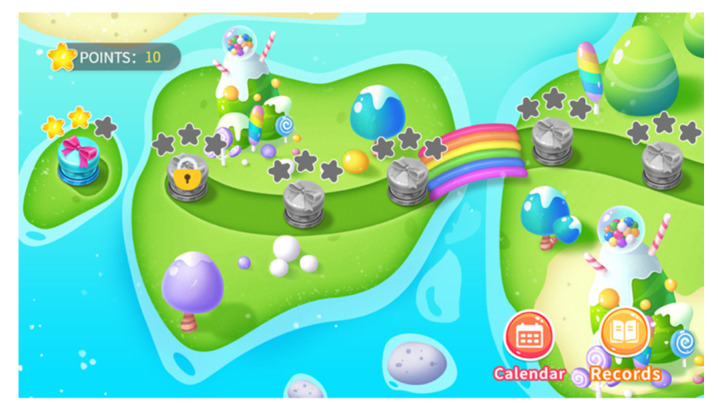
Screenshot of daily challenges within CareMom.

**Figure 2 ijerph-19-16824-f002:**
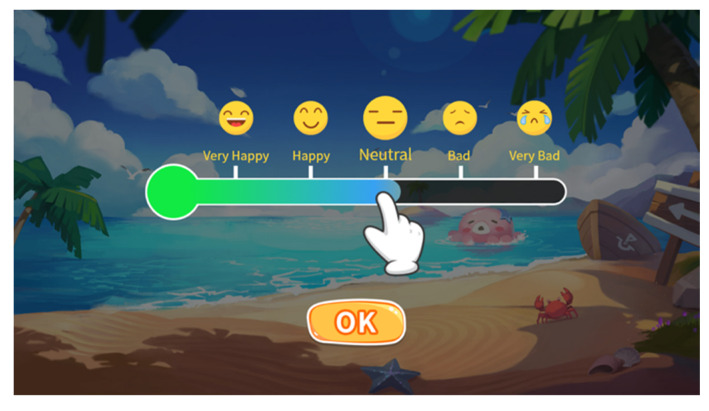
Screenshot of the mood tracker.

**Figure 3 ijerph-19-16824-f003:**
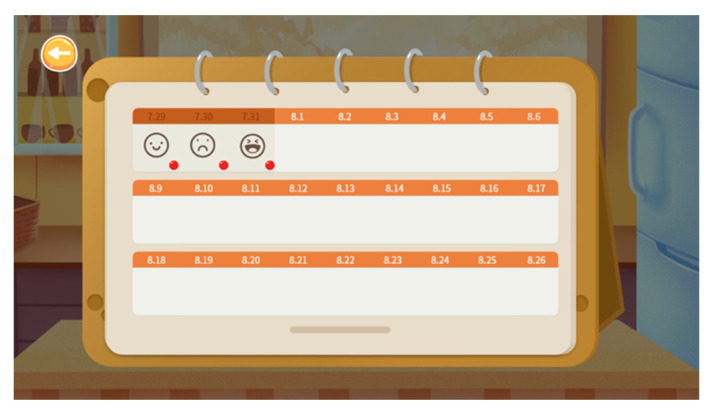
Screenshot of the mood calendar.

**Figure 4 ijerph-19-16824-f004:**
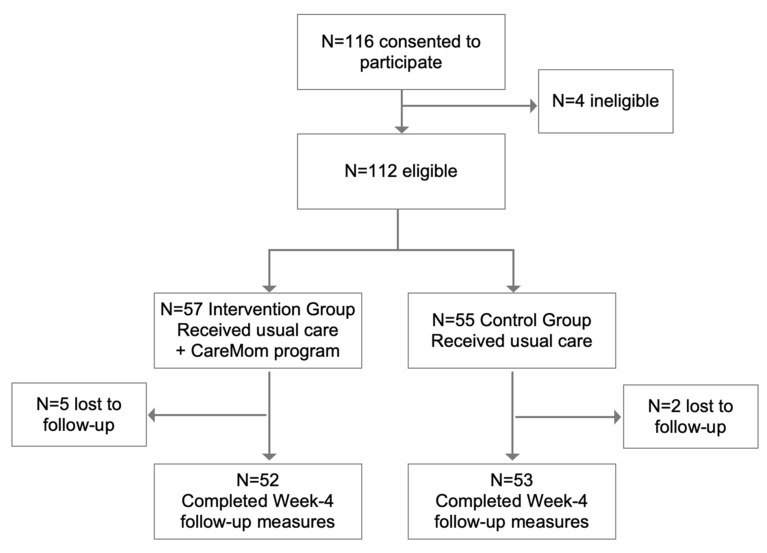
Participant recruitment flow.

**Figure 5 ijerph-19-16824-f005:**
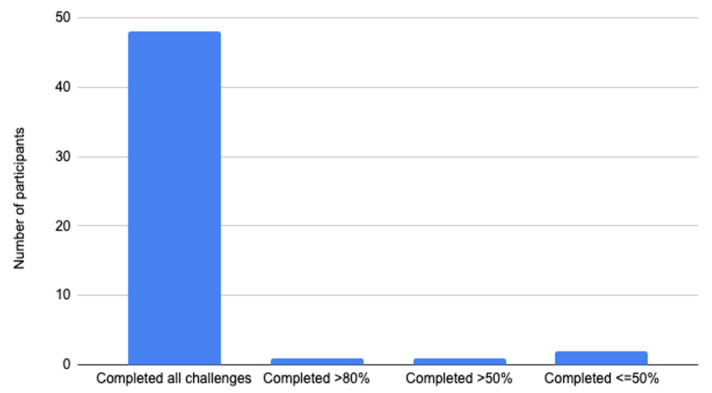
Participants’ engagement on the daily challenges.

**Table 1 ijerph-19-16824-t001:** Objectives of each daily challenge within the CareMom program.

Daily Challenge Number	Objectives of the Challenge
1	To introduce the structure and objectives of the CareMom program
2	Emotions in the postpartum period
3	Negative emotions of human beings, and the impacts of these emotions
4	Relationship between thoughts, emotions, and behaviors, and the importance of thoughts
5	Information about negative automatic thoughts, and how to recognize the negative automatic thoughts
6	Information about the “black and white thinking” cognitive distortion, and how to identify this cognitive distortion
7	Use the “examine the evidence” method to challenge cognitive distortions
8	Identify and challenge the “personalization” cognitive distortion
9	Identify and challenge the “mind reading” cognitive distortion
10	Identify and challenge the “overgeneralization” cognitive distortion
11	Identify and challenge the “labeling” cognitive distortion
12	Identify and challenge the “mental filtering” cognitive distortion
13	Information about the “should statements” cognitive distortion, and use the “semantic” method to challenge “should statement” cognitive distortions
14	Information about the “catastrophizing” cognitive distortion, and use the “decatastrophizing” method to challenge “catastrophizing” cognitive distortions
15–28	Quiz questions about the topics covered in the first 14 daily challenges

**Table 2 ijerph-19-16824-t002:** Demographic and clinical data of participants at baseline.

	CareMom (N = 57)	Control (N = 55)
Clinical data, Mean (SD)		
EPDS	4.54 (3.24)	5.42 (3.59)
GAD-7	1.74 (2.11)	2.26 (2.82)
Age, Mean (SD)	31.6 (3.35)	32.2 (3.89)
Delivery type, N (%)		
Natural	35 (61.4%)	25 (45.5%)
Caesarean section	22 (38.6%)	30 (54.5%)
Birth order, N (%)		
Primiparous	45 (78.9%)	42 (76.4%)
Multiparous	12 (21.1%)	13 (23.6%)

**Table 3 ijerph-19-16824-t003:** Mean EPDS and GAD-7 scores of the participants who completed the study, at baseline and weeks one to four.

	Baseline	Week 1	Week 2	Week 3	Week 4	
Mean EPDS (SD)						*p* ^(b)^
Control	5.42 (3.59)	4.77 (4.16)	5.06 (3.42)	4.77 (3.71)	4.55 (3.87)	0.12
CareMom	4.58 (3.37)	4.20 (4.11)	3.78 (3.50)	3.14 (3.03)	2.71 (2.75)	<0.001
*p* ^(a)^	0.44	0.48	0.26	0.10	0.037	
Mean GAD-7 (SD)						*p* ^(b)^
Control	2.27 (2.82)	1.72 (2.63)	2.02 (3.08)	1.85 (2.78)	1.88 (2.26)	0.27
CareMom	1.80 (2.11)	1.54 (2.22)	1.53 (2.27)	1.30 (1.85)	1.25 (2.12)	0.18
*p* ^(a)^	0.35	0.71	0.37	0.25	0.19	

*p*^(a)^: Bonferroni adjusted *p* values calculated via independent samples *t*-tests between the control and CareMom groups at week 4. *p*
^(b)^: Bonferroni adjusted *p* values calculated via paired samples *t*-tests between baseline and week 4 values.

**Table 4 ijerph-19-16824-t004:** Key results of the acceptability survey.

Question	Mean Score (SD)
What is the overall satisfaction level with the program?	4.58/5 (0.74)
To what extent would you recommend this program to your friends?	4.54/5 (0.80)
To what extent do you think the content of the program is related to your everyday life?	4.44/5 (0.62)
To what extent will you apply what you learnt from this program to your everyday life?	4.44/5 (0.58)
What is the overall satisfaction level with the program?	4.58/5 (0.74)

## Data Availability

Supporting data are available from Y.C. (drchenyixin@hotmail.com) on request.

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
