# Peer review of "Preventing Postpartum Depression in the Early Postpartum Period Using an App-Based Cognitive Behavioral Therapy Program: A Pilot Randomized Controlled Study"

_ijerph, 2022, doi:10.3390/ijerph192416824_

Round 1
Reviewer 1 Report
The manuscript entitled "Preventing Postpartum Depression in Early Postpartum Period Using an App-based Cognitive Behavioral Therapy Program: A Pilot Randomized Controlled Study" showed the effectiveness of postpartume app-based intervention on mental health. The results showed a significant reduction in depression scores at week four. This research discusses a very important topic and the manuscript is well-written. I have some comments below.
Major comments
1. Why authors used "pilot"? Please define the pilot study by citing literature, or delete the word "pilot".
2. The results of "within the group" were presented before "between groups" throughout the manuscript. The results of "between groups" were more important and should be presented first.
3. How did the authors decide the estimated effect size (d=0.6)? Please cite the literature.
4. The participants were recruited at the hospital. Did the researcher invite "all" the mothers who gave birth at that hospital during the study period? If so, please provide information about the total number of invited mothers and revise Figure 4, including the data about the number of mothers who did not consent to participate.
5. P.4 How did the authors decide on the contents? The contents seem to be mainly composed of cognitive restructuring, and not obtain mindfulness or other third-wave CBT. Please provide a rationale for the contents with evidence.
6. P.6 Please describe more in detail about the randomization process.
7. The survey was conducted every seven days for four weeks, but the authors used only the date of week 2 and week 4. Please state the analysis plan for the other data. Obtaining too much data can be a burden for participants and be an ethical issue.
8. P.10 Authors stated "To our knowledge, the present study is the first pilot study that....", but the relevant study has been published (universal intervention for preventing postpartum depression). Please consider the study and check the logic of the whole of your manuscript.
Nishi et al. The preventive effect of internet-based cognitive behavioral therapy for prevention of depression during pregnancy and in the postpartum period (iPDP): a large scale randomized controlled trial.
https://doi.org/10.1111/pcn.13458
9. Why such a high completion rate has been achieved? Please discuss more in the discussion, including the effect of the way of recruiting or system characteristics.
10. Did the authors think about the effect of COVID-19? If there is anything that may be affected by the COVID-19 pandemic, please provide short comments on the manuscript.
Minor comments
1. Inclusion criteria
The authors stated that "2) the mother was not diagnosed with any mental disorders." Please add the information of the period (when).
2. Trial registration and protocol
Has the protocol been registered in any clinical registration or published before?
3. Process
How do authors detect the completion rate? How do authors send reminders for learning? Does the program automatically send the message (contents and reminders) without professional personal support?
4. Table 3
The P value should be described exactly, without presenting >0.05.
Author Response
We really appreciate so many detailed comments and suggestions from the reviewer. The comments and suggestions do help us to refine a clearer scope of our manuscript. We have made substantial amendments of our manuscript, and a revised version of manuscript is submitted.
Major comments
- Why authors used "pilot"? Please define the pilot study by citing literature, or delete the word "pilot".
Response:
As described in the paper by Lowe (2019), a pilot study of interventions is for testing hypotheses regarding the effectiveness or efficacy of the intervention. In addition, a pilot study of interventions also helps to evaluate the acceptability of an intervention to its intended audience.
In the submitted paper, we describe a new interventional program. However, the program was not tested via a controlled study previously. Therefore, we define the study of this paper as a pilot study to evaluate: 1) the preliminary evidence of the effectiveness of the program; 2) the acceptability of the program. After this preliminary evaluation, we plan to enhance the program design and study design, so that we can test the program with a larger and more rigorous investigation in the future.
We add a citation and the explanation in the last paragraph of the Introduction section.
Reference:
Lowe, N. K. (2019). What is a pilot study?. Journal of Obstetric, Gynecologic & Neonatal Nursing, 48(2), 117-118.
- The results of "within the group" were presented before "between groups" throughout the manuscript. The results of "between groups" were more important and should be presented first.
Response:
We have now presented the results of “between groups” before “within the group”. We adjust the Abstract, Results and Discussion sections of the manuscript accordingly. Please see Abstract, and section 3.3 and 4.1.
- How did the authors decide the estimated effect size (d=0.6)? Please cite the literature.
Response:
Citations are added. Please see the second paragraph of Section 2.1.
- The participants were recruited at the hospital. Did the researcher invite "all" the mothers who gave birth at that hospital during the study period? If so, please provide information about the total number of invited mothers and revise Figure 4, including the data about the number of mothers who did not consent to participate.
Response:
During the recruitment process, the researchers of this study introduced the study procedures and requirements to mothers. If a mother voluntarily consented to participate in the study, they would be required to complete the consent and personal information forms. However, unfortunately, we did not track the information of the mothers who would not like to participate in the study.
However, we agree with the reviewer that the information of the mothers who did not consent to participate is important for better understanding the results of the program. Therefore, a sentence is added in the revised manuscript to remind readers to interpret the results with cautions. Please see the 4th paragraph of Section 4.2.
- 4 How did the authors decide on the contents? The contents seem to be mainly composed of cognitive restructuring, and not obtain mindfulness or other third-wave CBT. Please provide a rationale for the contents with evidence.
Response:
We have now added a rationale for the contents in the second and third paragraphs of Section 2.3.
- 6 Please describe more in detail about the randomization process.
Response:
The detailed randomization process is added in the first paragraph of Section 2.4.
- The survey was conducted every seven days for four weeks, but the authors used only the date of week 2 and week 4. Please state the analysis plan for the other data. Obtaining too much data can be a burden for participants and be an ethical issue.
Response:
The results of week 1 and week 3 have been added in Table 3.
In the fifth edition of the Diagnostic and Statistical Manual of Mental Disorders, the postpartum period is defined as being within 4 weeks postpartum. To learn more about the depressive and anxiety symptoms of new mothers, we conducted the surveys weekly. The detailed data might be helpful for other researchers to better understand the emotional state of new mothers at their early postpartum period.
- 10 Authors stated "To our knowledge, the present study is the first pilot study that....", but the relevant study has been published (universal intervention for preventing postpartum depression). Please consider the study and check the logic of the whole of your manuscript.
Nishi et al. The preventive effect of internet-based cognitive behavioral therapy for prevention of depression during pregnancy and in the postpartum period (iPDP): a large scale randomized controlled trial. https://doi.org/10.1111/pcn.13458
Response:
We have added the study in the manuscript, and the logic of the whole manuscript has been adjusted accordingly. Please see the adjustments in the Introduction and Discussion sections.
- Why such a high completion rate has been achieved? Please discuss more in the discussion, including the effect of the way of recruiting or system characteristics.
Response:
We think the design of the CareMom program was one of main factors for the high completion rate. Other factors might include the way of recruitment and the way of sending reminders. We have now added some discussion about the completion rate in Section 4.2.
- Did the authors think about the effect of COVID-19? If there is anything that may be affected by the COVID-19 pandemic, please provide short comments on the manuscript.
Response:
We think the COVID-19 pandemic had little influence on this study, because the intervention was provided via the online program and the intervention heavily relied on the cognitive restructuring. However, we think if this program heavily relies on behavioral interventions, the study might be influenced by COVID-19 pandemic.
Minor comments
- Inclusion criteria
The authors stated that "2) the mother was not diagnosed with any mental disorders." Please add the information of the period (when).
Response:
The information of the period has been added in the revised manuscript.
- Trial registration and protocol
Has the protocol been registered in any clinical registration or published before?
Response:
The protocol has not been published before. This study was defined as a pilot study and aimed to explore the acceptability and preliminary effectiveness of the program. Therefore, we did not register the study with clinical trial. After this pilot study, we might refine the product and conduct a registered clinical trial study to further examine the effectiveness of the program.
- Process
How do authors detect the completion rate? How do authors send reminders for learning? Does the program automatically send the message (contents and reminders) without professional personal support?
Response:
We have now clarified the process in the last paragraph of Section 2.4.
In short, the reminders were not automatically sent by the program. They were sent by the study administrator manually via the WeChat program. The study administrator did not provide any professional/personal support to the participants.
- Table 3
The P value should be described exactly, without presenting >0.05.
Response:
The exact P values have been added in Table 3.
Reviewer 2 Report
The study was well designed and the manuscript was well written, the discussion section should be rewritten to achieve better understanding. In the discussion section, the authors simply repeat the results of previous studies and their own findings. However, authors should discuss possible reasons for their findings and discrepancies between their findings and the literature. In addition, they should make recommendations based on their findings.
Considering the issues investigated and the results observed in the aforementioned paper, I believe that this submission meets the criteria published in the International Journal of Environmental Research and Public Health following the major changes I have suggested.
Author Response
We really appreciate the comments and suggestions from the reviewer. We have made substantial amendments of our manuscript, and a revised version of manuscript is submitted.
The study was well designed and the manuscript was well written, the discussion section should be rewritten to achieve better understanding. In the discussion section, the authors simply repeat the results of previous studies and their own findings. However, authors should discuss possible reasons for their findings and discrepancies between their findings and the literature. In addition, they should make recommendations based on their findings.
Response:
We have refined the discussion section. We discussed the findings and the discrepancies between our findings and the literature about the preliminary effectiveness and acceptability of CareMom program. Please see the section 4.1 and 4.2.
Round 2
Reviewer 1 Report
I think that the revised manuscript achieved the level of acceptable publication in IJERPH after minor revisions on the point I required to address.
Author Response
We really appreciate the quick feedback from the reviewer. We have addressed the suggestion from the reviewer in the revised manuscript.
- The revised manuscript has been much improved. I have one last comment: the statement that there was no published protocol and clinical trial registration should be described in the limitation section to make it easy for readers to assess the quality of this intervention study.
Response:
We have now added a statement in the limitation section to remind readers to interpret the results of this paper with cautions. Please see Section 4.3.